# Investigation of Transport Mechanism and Nanostructure of Nylon-6,6/PVA Blend Polymers

**DOI:** 10.3390/polym15010107

**Published:** 2022-12-27

**Authors:** Hamdy F. M. Mohamed, Esam E. Abdel-Hady, Wael M. Mohammed

**Affiliations:** Physics Department, Faculty of Science, Minia University, Minia P.O. Box 61519, Egypt

**Keywords:** polymer blend, positron annihilation, free volume, nylon-6,6, poly(vinyl alcohol), proton conductivity, thermogravimetric analysis, wide-angle X-ray diffraction

## Abstract

A casting technique was used to prepare poly(vinyl alcohol) (PVA) blend polymers with different concentrations of Nylon-6,6 to increase the free-volume size and control the ionic conductivity of the blended polymers. The thermal activation energy for some blends is lower than that of pure polymers, indicating that their thermal stability is somewhere in between that of pure Nylon-6,6 and pure PVA. The degree of crystallinity of the blend sample (25.7%) was lower than that of the pure components (41.0 and 31.6% for pure Nylon-6,6 and PVA, respectively). The dielectric properties of the blended samples were investigated for different frequencies (50 Hz–5 MHz). The σ_ac_ versus frequency was found to obey Jonscher’s universal power law. The calculated values of the s parameter were increased from 0.53 to 0.783 for 0 and 100 wt.% Nylon-6,6, respectively, and values less than 1 indicate the hopping conduction mechanism. The barrier height (Wm) was found to increase from 0.33 to 0.72 for 0 and 100 wt.% Nylon-6,6, respectively. The ionic conductivity decreases as the concentration of Nylon-6,6 is blended into PVA because increasing the Nylon-6,6 concentration reduces the number of mobile charge carriers. Positron annihilation lifetime (PAL) spectroscopy was used to investigate the free volume’s nanostructure. The hole volume size grows exponentially with the concentration of Nylon-6,6 mixed with PVA. The Nylon-6,6/PVA blends’ free-volume distribution indicates that there is no phase separation in the blended samples. Mixing PVA and Nylon-6,6 resulted in a negative deviation (miscible blends), as evidenced by the interaction parameter’s negative value. The strong correlation between the free-volume size and other macroscopic properties like ionic conductivity suggests that the free-volume size influences these macroscopic properties.

## 1. Introduction

Nylon was one of the early polymers developed by Carothers [1]. The term “Nylon” is a generic designation for synthetic polymers known generically as aliphatic polyamides, and its other known representatives are Nylon-6,6, Nylon-6, Nylon-6,9, Nylon-6,10, Nylon-6,12, Nylon-11, Nylon-12, and Nylon-4,6 [2]. Nylon is a crystalline polymer with high modulus, strength, impact properties, low coefficient of friction, and resistance to abrasion [3]. Although the materials possess a wide range of properties, polyamides all contain the amide (-CONH-) linkage in their backbone. Polyamide is the most widely used thermoplastic polymer among engineering plastics due to its superior mechanical properties and chemical resistance [3]. Nylon-6 has traditionally been used as a material substitute for natural fibres [4,5], but it is increasingly being used in auto parts because of its excellent resistance to obstruction and scraped spots. Nylon-6 auto parts are typically handled through melt processing, for example, expulsion or infusion shaping, and chain extenders can be used to present branches and increase the subatomic weight [6,7]. Diverse reinforcing materials can also be introduced to prepare Nylon-6,6 composites [8,9].

Poly(vinyl alcohol) (PVA) is well known for its low cost, excellent transparency, toughness, biodegradability, and gas-barrier properties. It has been widely used as a textile sizing and finishing agent, in photosensitive coatings and food packaging, and in adhesives for paper, wood, textiles, and cowhide [10]. Be that as it may, PVA can only occasionally be utilized as a thermoplastic polymer because of its unfortunate warm strength and high dampness retention. PVA can, however, be utilized as a modifier added to other viable polymers [11,12,13], for example, essential polyamides, through liquefied mixing cycles to further develop gas boundary and biodegradability, because mix readiness is generally the most catalytic and financially feasible choice for changing polymer properties. It is notable that polyamides have great boundary properties against hydrocarbon solvents (e.g., xylene, toluene, white soul, and fuel) and great bonding to various resinous substrates or coatings.

When melt-blending 25 wt.% PVA with Nylon-6, Koulouri and Kallitsis [14] proposed that these polymers are compatible in this composition because only one dynamic glass transition temperature was found using dynamic mechanical analysis (DMA). Furthermore, an examination of the properties of blends of Nylon-6 and PVA with fluctuating levels of hydrolysis was conducted [12,13]. These outcomes support the possibility that PVA is somewhat miscible with Nylon-6 at the subatomic level when the PVA content is equivalent or not exactly equivalent to the compared basic PVA content. Reinforcing materials are often used for compounding with polymeric materials to improve their mechanical or thermal properties. According to some studies conducted to improve the physical properties of polymer–clay nanocomposites, a small amount of clay was added to a Nylon-6–clay nanocomposite, and various improvements were achieved: higher polymer strength, higher heat resistance, enhanced thermal stability, enhanced gas-barrier properties, and so on [15]. The miscibility of homopolymer/copolymer blends has been successfully described using the binary interaction model. Hydrogen bonds, ionic bonds, and dipole–dipole interactions are the most frequent types of specific intermolecular interactions that take place between two separate polymer chains [16].

The studies of the dielectric complex, the conduction mechanism, and the barrier height for the blended samples in a wide range of frequencies (50 Hz to 5 MHz) have been carried out to investigate the dielectric constant of the samples, which reveals the polarization in the samples and their ability to store electrical energy, and to identify the type of conduction mechanism. The conduction mechanism of PVA/TiO_2_ was investigated by Khairy et al. [17], and the correlated barrier hopping mechanism was the major conduction mechanism.

Free volume is a significant boundary that portrays polymer properties. The free volume idea has been effectively applied in polymer material science to portray different properties of polymers like versatility, consistency, and so on. The quantitative measurement of free-volume size and number has grown in popularity and significance. The free-volume size determination can be performed using the positron annihilation lifetime (PAL) technique. It is a non-destructive method with a minimum free-volume size determination of around 1 Å. The positron annihilation lifetime fundamentals will be discussed in the second section. Li et al. [18] concentrated on the free-volume varieties of Nylon-12/PVA melt-blends treated in supercritical carbon dioxide (ScCO_2_) using PAL spectroscopy. They found that the free volume opening size and its items declined with the increment in PVA content. The free-volume holes in Nylon-12/PVA melt-blends expand to varying degrees after 1 h in ScCO_2_ at 50 °C and 20 MPa, depending on the weight proportion of PVA. After delivering CO_2_, the free-volume size diminishes as a component of the passed time because of subatomic unwinding, and the unwinding time steadily relies upon the substance of PVA in the melt-blends.

In the present investigation, PVA polymers were blended with Nylon-6,6 using the solvent method to yield a high-barrier electrical conductivity material. This study’s main objective is to explore the electrical properties, free volume, and mixing properties of PVA with varying Nylon-6,6 content to conform to the PVA/Nylon-6,6 blend. To figure out the electrical properties and free-volume characteristics of the prepared PVA/Nylon-6,6 blend films, thermogravimetric analysis (TGA), wide-angle X-ray diffraction (WAXD), electrical conductivity measurements, and PAL spectroscopy were performed. Potential reasons for the significantly superior conductivity hindrance and free-volume properties of the PVA/Nylon-6,6 blended films are proposed.

## 2. Positron Annihilation Spectroscopy

The positron annihilation technique [19,20,21,22,23,24,25,26,27,28,29,30,31,32,33] has been developed into a strong portrayal device for the investigation of free volume size and free volume distribution in polymeric material. Estimates of the lifetimes of the positrons can lead to a genuinely precise estimate of the free volume of the angstrom (2–10 Å) range. At the point when positrons are embedded in a polymeric material, some of them cooperate with the electron mists in the material and demolish the two gamma beams of 511 keV. Be that as it may, a small number of vivacious positrons, embedded in a polymer as a rule, form the bound state with an electron, positronium (Ps). The singlet state is *para*-positronium (*p*-Ps) with the spins of the positron and electron being antiparallel, and the triplet state is *ortho*-positronium (*o*-Ps) with parallel spins. The *p*-Ps and *o*-Ps are framed at a proportion of 1 to 3. The *p*-Ps are destroyed into two demolition gammas (each of 511 keV) with a long period of 125 ps, while the positron in *o*-Ps likewise demolishes with one of the encompassing electrons by means of a cycle called “pick-off annihilation”, prompting two demolition gammas (each of 511 keV as well). The lifetime of the *o*-Ps after take-out demolition in polymers is completed from 1 to 5 ns, which is emphatically reliant upon the size of sub-nanometer scale openings, where Ps is confined. The pick-off annihilation lifetime is a lot more modest than the self-annihilation lifetime of *o*-Ps (~142 ns). Positrons that are not framed Ps go through two demolition gammas (each of 511 keV) after thermalization with one of the encompassing twists including inverse electrons with a long period of 400−500 ps.

The positron lifetime range is traditionally portrayed using a number of discrete exponentials [34]. The exploratory information *y*(*t*) is communicated as a tangled articulation (by an image *) of the PAL system resolution function *R*(*t*) and a limited number (n) of negative exponentials [35]:(1)$$y(t)=R(t)*\left[N_{i}\sum_{i=1}^{n}(I_{i}/\tau_{i})\exp(-t/\tau_{i})+B\right]\;,$$
where *N_i_* is the standardized counts and *B* is the background, each with a trademark positron lifetime of *τ_i_* with a relative intensity of *I_i_* (∑*I_i_* = 1). From non-straight least-squares fits, the mean lifetimes *τ_i_* (*i* = 1–3) and their intensities *I_i_*, the zero time of the spectrum, and the spectrum background *B* have been determined. Each positron lifetime spectrum containing a complete count of 3.0 × 10^6^ counts was dissected utilizing the PALSfit3 program [36], and source rectification was applied in the data examination. The analysis showed that the best variance ratio (χ^2^ < 1.1) and most acceptable standard deviations were obtained when each spectrum is fitted in terms of three-lifetime components, and the shortest lifetime component *τ*_1_ was set to be constant (125 ps) and its intensity *I*_1_ was set as a guess value.

On the off chance that the veritable positron lifetime values include a greater constant flow (continuous distribution) than the discrete nature considered, the positron annihilation lifetime range might be depicted by a consistent rot of the accompanying structure [37],
(2)$$y(t)=R(t)*\left[N_{t}\int_{0}^{\infty }(I(\tau )/\tau )\exp(-t/\tau )d\tau +B\right]\;,$$
where
(3)$$\int_{0}^{\infty }I\left(\tau \right)d\tau =1.$$

Furthermore, *N_t_* is the absolute count of the PAL spectrum [31]. For the regular discrete term investigation, the routine LT10.0 program [19,38,39,40] has been utilized. In this examination, the model capability Equation (2) accepting the number of terms in the range is fitted to the trial data of interest. Using the LT10 program, typical conveyance is accepted for *o*-Ps annihilation rate λ which is equivalent to 1/*τ*_3_.

The component associated with the annihilation of the *o*-Ps state is generated by pick-off with the encompassing electrons. The *o*-Ps lifetime is supposed to rely upon the free-volume hole sizes in polymer materials because the annihilation rate λ_3_ (λ_3_ = 1/*τ*_3_) of *o*-Ps is relative to the crossover of the positron and pick-off electron wave capabilities. A connection between the free volume in subatomic frameworks and the noticed *o*-Ps lifetimes has been figured out from the positronium hole theory [41]. This gives quantitative data about the free-volume size from the PAL spectrum. The average of the *o*-Ps hole volume radius (*R*), which was tested using the *o*-Ps lifetime (*τ*_3_), was determined from the accompanying connection [42,43]:(4)$$\tau_{3}=0.5{\left\{1-\frac{R}{R_{o}}+\frac{1}{2\pi }\sin\left(\frac{2\pi R}{R_{o}}\right)\right\}}^{-1}(\mathrm{ns}),\;$$
where *R_o_* = *R* + Δ*R* and Δ*R* = 0.1656 nm is the thickness of the homogenous electron layer in which the positron destroys [44]. The *o*-Ps hole volume size *V*_f_ in nm^3^ is given as:*V*_f_ = 4 *πR*^3^/3(5)
so that the hole volume distribution is expressed as:(6)$$\alpha_{3}(V)={-(2\pi )}^{-0.5}\sigma_{3}^{-1}\exp\{-\frac{{\left[Ln(\lambda )-Ln(\lambda_{3o})\right]}^{2}}{2\sigma_{3}^{2}}\}\lambda^{-1}d\lambda \frac{dV}{d\lambda },$$
where
(7)$$\frac{dV}{d\lambda }=-\frac{4\pi R^{2}{\left(R+R_{o}\right)}^{2}}{2R_{o}}{\left\{1-\cos\left(\frac{2\pi R}{R+R_{o}}\right)\right\}}^{-1},$$ as derived from Equations (4) and (5).

According to the Ps-hole theory, the probability of *o*-Ps formation is related to the fractional of the *o*-Ps hole volume *f* (%) in a polymer. A simple semi-empirical equation has been proposed [45]:*f* = *C I*_3_ *V*_f_, (8)
where *C* is a parameter (for the particular material) to be determined. For convenience, the relative fractional of the *o*-Ps hole volume is defined as [46]:*f*_r_ = *I*_3_ *V*_f_. (9)

## 3. Materials and Methods

The poly(vinyl alcohol) (PVA) with a molecular weight (*M*_w_) of 72,000 g/mol was purchased from MERCK-Schuchardt, Germany. Nylon-6,6 films were purchased from Goodfellow, Japan, while formic acid was provided from Sigma Aldrich, Germany, as a solvent. The compound design of the despise polymers is displayed in Figure 1. Nylon-6,6 and PVA were dissolved in formic acid with adjustments of the concentration of Nylon-6,6 in PVA to 0, 20, 40, 60, 80, and 100 wt.%. The polymer blend solution was prepared at 70 °C with a guiding season of 24 h. The blend solution was projected onto a Teflon Petri dish, and the polymer blend film was eliminated in the wake of drying at room temperature (25 °C). The thickness of the blended film is from 103 to 142 μm.

The weight reduction’s temperature dependence was examined to determine the membranes’ thermal stability. Using Q50, USA, and a heating rate of 10 °C/min in a N_2_ inert gas environment, thermogravimetric analysis (TGA) measurements were carried out on all samples with an area of 1 × 1 cm^2^. Additionally, the TGA results were used to calculate a derivative thermogravimetric (DTG) analysis for each membrane.

The crystal structure of the prepared samples understudy was investigated using a wide-angle X-ray diffraction (WAXD) technique on a Bruker (D8) diffractometer at 30 kV and 30 mA using a parabolic multilayer membrane mirror with CuK (=1.54184 Å) incident radiation. Scattering profiles were recorded with a scintillation one-dimension position-sensitive detector over a range of 5° < 2θ < 80° with a step of 0.05° and at a rate of 0.5°/min.

An LCR meter (Hioki 3532, Japan) was used to measure the many parameters such as impedance, parallel-equivalent static capacitance, loss coefficient, and conductance for all samples. Using some of these parameters, the dielectric properties and complex impedance were calculated as a function of frequencies from 50 Hz to 5MHz. The measurements were performed under a vacuum at 25 °C. The out-of-plane value of the AC ionic conductivity *σ_AC_* can be acquired using the conductance G as [47]:(10)$$\mathrm{Ionic}\;\mathrm{conductivity}\;\sigma_{AC}=\frac{L\;G\;}{\;A}.$$
where *L* and *A* are the film thickness and the cross-sectional region of the electrodes, respectively.

The PAL method makes use of the ^22^Na radioisotope, a positron source with activity 20 μCi. The positron source was prepared by coating a Kapton foil that was indistinguishable after drying with 3–5 drops of a carrier-free ^22^NaCl arrangement (7 μm thick, 1 cm^2^ area). The Kapton absorbed about 10% of the positrons that came from the source. The positron source retention was removed during the PAL spectra examination. Two comparable films with a volume of around 1 × 1 × 0.1 cm^3^ surrounded the positron source. After that, the sample/positron source/sample sandwich was placed in the moisture holder between the two detectors of the PAL spectrometer (fast–fast coincidence system). According to all accounts, Kapton is the main polymer with no positron yield, i.e., it lacks a long-lived component [48]. The time resolution was 240 ps (full width at half maximum, FWHM) as determined using the PALsfit3 program [49]. The current PAL technique was portrayed using the standard examples of synthetic fused silica (NMIJ CRM 5601-a) and polycarbonate (NMIJ CRM 5602-a) provided by the National Metrology Institute of Japan, National Institute of Advanced Industrial Science and Technology (AIST) [50,51]. The standard examples were expected for use in controlling the accuracy of the estimated information or approving the estimation conditions, and they obtained results using the positron annihilation lifetime method for polymers and protectors having a positronium part with a lifetime longer than 1 ns. The sample/positron source/sample sandwich was placed in a glass cylinder to play out the PAL estimations in a vacuum at room temperature (around 25 °C).

## 4. Results and Discussion

The thermogravimetric analysis (TGA) estimations shown in Figure 2A are used to perform the thermal analysis of the prepared samples. The derivative thermogravimetric (DTG) examination, which is depicted in Figure 2B, was used to break down the TGA spectra for each of the blended samples. The samples can be thermally degraded for different reasons, such as losing the adsorbed water, elimination of any equivalent groups or hydrogen atoms, or disposal of carbon by converting it into carbon dioxide, as was introduced in recently published studies [52,53,54]. In the present case, the TGA or/and DTG figures show that weight loss practices can be broken down into four areas. The temperatures in regions I and II range from 30 to 160 °C and from 160 to 340 °C, respectively. In addition, the temperature in region III ranges from 340 to 520 °C while that in region IV is above 520 °C. For pure Nylon-6,6, weight loss was observed in three steps as regions I+II, III, and IV, while it was observed in regions I, II, and III + IV for pure PVA, as shown in the derivative weight curve. The weight loss in temperature region I and regions I + II is related to the dehydration of PVA and Nylon-6,6, respectively. The decomposition due to the degradation of PVA, which represents the major weight loss of PVA, was observed in region II, while the decomposition due to the degradation of Nylon-6,6 took place in region III. The third stage of decomposition for PVA and Nylon-6,6 took place in regions III+IV and IV, respectively. This is due to the cleavage of the PVA [55,56,57,58] and Nylon-6,6 backbone [59].

In general, for all samples, the influence of the Nylon-6,6 concentration on the weight loss of the samples can be easily observed. This is clear in regions II and III where the peak shifts gradually to a lower temperature by increasing the ratio of Nylon-6,6. The addition of Nylon-6,6 led to a decrease in the temperature of the decomposition of PVA from 263 to 243 °C, and the addition of PVA led to a decrease in the temperature of the decomposition of Nylon-6,6 from 450 to 411 °C. Therefore, increasing the concentration of Nylon-6,6 leads to a decrease in the melting temperature for both PVA and Nylon-6,6. A decrease in the liquefying guide in a polymeric blend can be expected to have morphological effects (decrease in lamellar thickness) as well as thermodynamic effects (polymer connections) [16]. In conclusion, the thermal stability of the blended samples has been clearly influenced by adding Nylon-6,6 to PVA.

The prepared blend polymers have two decomposition temperatures from the TGA spectra, whereas the pure polymer only has one. Calculating the activation energy of thermal decomposition *E_b_* for each decomposition temperature is fascinating. The Horowitz and Metzger connection was used to evaluate the activation energy of thermal decomposition *E_b_* [60]:(11)$$\ln\left[\ln\left(\frac{W_{o}-W_{f}}{W_{t}-W_{f}}\right)\right]=\frac{E_{b}△T}{GT_{s}^{2}}.\;$$
where *W*_o_ is the weight of the sample at onset temperature, *W*_f_ is the weight of the sample at the endset temperature, *W*_t_ is the excess sample weight at temperature *T*, and *G* is characterized as the gas constant. Δ*T* is equivalent to *T* − *T*_s_, where *T*_s_ is the reference when:(12)$$\frac{W_{o}-W_{f}}{W_{t}-W_{f}}=e=2.718$$

*E*_b_ is calculated for each sample and recorded in Table 1 by determining the slope of the linear function between $\ln\left[\ln\left(\frac{W_{o}-W_{f}}{W-W_{f}}\right)\right]$ and Δ*T*. As can be seen from Table 1, pure PVA has a higher *E*_b_ than pure Nylon-6,6, indicating that PVA is more thermally stable than Nylon-6,6. On the other hand, *E*_b_ for some blends is lower than that of pure polymers, indicating that their thermal stability is somewhere in between that of Nylon-6,6 and PVA.

The WAXD for the pure PVA, pure Nylon-6,6, and blended polymers (60 wt.% Nylon-6,6/40 wt.% PVA) is shown in Figure 3. It is clear from Figure 3 that the WAXD pattern for the pure PVA contains two main diffraction peaks, which were observed at 2*θ* = 19.5° and 41°. The current findings are consistent with those of Gupta et al. [61]. The first peak corresponds to the reflection plane of (101), while the second one indicates the crystalline phase of PVA. For pure Nylon-6,6, the WAXD pattern shows that Nylon-6,6 has a polymorphic structure (crystalline and amorphous). This mixed structure was identified with the two main diffraction peaks; 2*θ* = 20.4° associated with (200) planes, and 2*θ* = 23.91° associated with (002/202) plants of crystal form α [62].

In addition, these main diffraction peaks reveal the monoclinic phase and agree with the data of Farias-Aguilar et al. [63] and Bunn et al. [64]. For the blend sample (60 wt.% Nylon-6,6 and 40 wt.% PVA), the WAXD pattern shows all the peaks of Nylon-6,6 and PVA. One can easily observe an increase in the intensity of the peak at 2*θ* about 20°, which is due to the overlap between the PVA peak and the Nylon-6,6 peaks. The concentration of Nylon-6,6 to PVA for this sample was confirmed by calculating the ratio of the intensity of the normalized peaks as follows: the ratio between the intensity of the two normalized peaks of the pure Nylon-6,6 is 8100/10,800 = 0.75 and that of the blended sample is 7100/3800 = 1.87; the difference between the two obtained values is 1.87 − 0.75 = 1.12, which represents the ratio of the intensity of PVA in the overlapped peak; then the concentration of PVA in the blended sample can be obtained with the ratio of 1.12/1.87 = 0.6 or 60 wt.%, which exactly agrees with the experimental data.

It is interesting to calculate the degree of crystallinity of the samples using the WAXD pattern and Fityk program [65]. The mean peaks of the pattern were deconvoluted into crystalline and amorphous peaks as blue- and red-colored peaks, respectively, as shown in Figure 3. The sharper peaks reveal the crystalline part of the sample, which corresponds to the intermolecular hydrogen bonds of PVA, and the wider peak indicates the amorphous part of the PVA film [66,67,68]. The degree of the crystallinity *X*_c_ can be determined using the equation:(13)$$X_{c}\left(\%\right)=\frac{Area\;under\;the\;sharper\;sub-peaks}{Total\;area\;under\;the\;main\;peaks}$$

Pure Nylon-6,6 and pure PVA have degrees of crystallinity *X_c_* of 41.0 and 31.6%, respectively. Additionally, the blended sample’s crystallinity, which is approximately 25.7%, can be determined in the same manner. As a result, the degree of crystallinity of the blend suffers as a result of the negative effects of each component. Additionally, the degree of crystallinity of the pure Nylon-6,6 decreased from 41.0 to 25.7% for the blended Nylon-6,6. Cagiao et al.’s [69] findings are supported by the current samples’ degree of crystallinity behaviour.

The grain size *D* was calculated by applying the Scherrer equation [70]:(14)$$D=\frac{0.9\lambda }{\beta \cos\left(\theta \right)}\;.$$
where *λ* is the wavelength and *β* is the full width at half maximum (FWHM). Table 2 lists the peak position, FWHM, and grain size for the crystalline and amorphous peaks. Table 2 makes it abundantly clear that the blend sample’s degree of crystallinity is lower than that of the homopolymers due to the blend sample’s smaller grain size (2.82 nm). In addition, the TGA data show that each component has a lower melting point (the temperature at which it begins to decompose) than homopolymers. 

The dielectric constant reflects the ability of any material to store electrical energy, which is significantly related to the dipole polarization resulting from the influence of an external electric field [71,72]. Figure 4A shows the frequency dependence of the dielectric constant *ε*^/^ for the present samples. The figure shows that the dielectric constant irregularly decreases with frequency. Such behaviour follows the Maxwell–Wagner interfacial polarization model [73]. Moreover, the low frequencies have the greatest effect *ε*^/^ because the dipoles have enough time for interfacial polarization. However, at high frequencies, there is not enough time for the dipoles to be oriented in the direction of the applied field. Pure PVA had the highest dielectric permittivity, which could be attributed to its polar nature [74]. Therefore, embedding Nylon-6,6 into PVA decreases the dielectric constant, which improves its ability to store electric energy. The frequency-dependent dielectric loss *ε*^//^ for the current samples is depicted in Figure 4B. Similar to the depicted behaviour of the dielectric constant with frequency, it was observed that the dielectric loss decreased rapidly and remained constant at higher frequencies. This behaviour conforms to the logical theory of Koop’s phenomena and the Maxwell–Wanger interfacial polarization model [75]. This reveals that grain and grain boundary make up the dielectric materials in multilayer capacitors. Highly conductive grains have a significant impact at high frequencies, whereas grain boundaries with poor conductivity at low frequencies are more effective [76].

The movement of charge carriers through the polymer’s chain is known to cause a sample’s AC ionic conductivity to develop [77]. Figure 5 illustrates the relationship between ln (*ω*) and ln (*σ_ac_*) for pure PVA and PVA blended with various concentrations of Nylon-6,6, which were obtained using Equation (10). As shown in Figure 5, *σ_ac_* decreases as the concentration of Nylon-6,6 increases. The maximum AC ionic conductivity *σ_ac_* is 1.25 × 10^−4^ S/m for pure PVA, which is considered one of the highest values in the literature [78]. Two regions are observed in the figure with different slopes. At low frequencies, the first region (I) reflects the dc conductivity which occurs as a result of the charge carrier’s displacement. Region (II) represents the dispersion region due to the ac conduction. The ac conductivity *σ_ac_* obeys Jonscher’s universal power law, which is described as [47]:(15)$$\sigma_{ac}\;\left(\omega \right)=A\omega^{S}.$$
where *A* is a temperature-dependent constant that identifies the polarizability strength, *ω* is the angular frequency, and *S* is the frequency exponent with values 0 < *S* < 1. The calculated value of *S* increased with an increase in the Nylon-6,6 concentration; it is 0.53, 0.68, 0.77, 0.78, 0.782, and 0.783 for 0, 20, 40, 60, 80, and 100 wt.% Nylon-6,6 concentration blended with PVA, respectively. The *S* values are less than 1 for all the blended samples, which indicates that the conduction mechanism is a hopping mechanism. Such behaviour was obtained for PVA1-xNaIx in a previous study conducted by Hmamm et al. [47].

The energy required to remove the electrons from their sites to infinity is called barrier height *W_m_*. It can be calculated by the relation [79]:(16)$$W_{m}=\frac{6K_{B}T}{1-S}.$$
where $K_{B}$ and T are the Boltzmann constant and the temperature, respectively. For Nylon-6,6 concentrations of 0, 20, 40, 60, 80, and 100 wt.% blended with PVA, the calculated values of *W_m_* are 0.33, 0.48, 0.67, 0.71, 0.71, and 0.72 eV, respectively. With increasing Nylon-6,6 concatenations blended with PVA, the value of *W_m_* increases, indicating a decrease in the number of charge carriers able to leap over the barrier height. This provides an explanation for the decrease in AC conductivity caused by increasing Nylon-6,6 concentrations.

Complex impedance analysis makes it simple to describe the interactions that take place at the grain and grain boundary. The frequency dependence of the real and imaginary impedances (Z^/^, Z^//^) for the prepared samples is depicted in Figure 6. For the real impedance Z^/^ (Figure 6A), the maximum value is observed for the pure Nylon-6,6 and decreases gradually with increasing PVA content, which indicates that PVA addition improves the conductivity of the prepared samples. Furthermore, it is observed that the variation in Z^/^ in the low frequency and its saturation in the high-frequency range suggest a different polarization behaviour in the samples, such as orientation and electronic polarization [80]. Figure 7A shows the real impedance Z^/^, where the maximum value for pure Nylon-6,6 decreases gradually as the PVA content rises, indicating that the prepared samples’ conductivity is enhanced by PVA addition. In addition, it is observed that the low-frequency variation of Z^/^ and its saturation in the high-frequency range suggest a distinct orientation and electronic polarization behaviour in the samples [80]. An asymmetry and broad peaks are visible in the imaginary impedance Z^//^ with frequency [Figure 6B]; it shows multiple relaxations and a deviation from the Debye behaviour. Such characteristics are known in semiconductors [75].

The Cole–Cole spectra of the synthesized samples are shown in Figure 7A. The diameter of the semicircle or the intercept of the semicircle with the Z^/^-axis increases with decreasing Nylon-6,6 content in the samples. The diameter of the semicircle or the intercept of the semicircle with the Z^/^-axis can be defined as the resistance of the sample (*R_s_*). Then, the DC ionic conductivity was calculated as follows:(17)$$DC\;{\mathrm{ionic}\;\mathrm{conductivity}\;\sigma }_{DC}=\frac{L\;}{\;R_{s}\;A}.$$

The DC ionic conductivity σ*_DC_* is depicted in Figure 7B as a function of the proportion of Nylon-6,6 to PVA. The DC ionic conductivity clearly decreases as the ratio of Nylon-6,6 to PVA polymer increases. This could be because increasing the Nylon-6,6 ratio reduces the number of mobile charge carriers.

Utilizing source corrections and three-lifetime components, the PALsfits3 [36] analysis of the positron annihilation lifetime spectra for the current samples yielded the best variance ratio, χ^2^ (<1.1). The annihilation of *p*-Ps atoms has the shortest lifetime component *τ*_1_ (=0.125 ns), while the free annihilation of positrons in the polymer matrix has the intermediate lifetime component *τ*_2_ (0.427–0.618 ns). The longest-lived component *τ*_3_ is attributed to the *o*-Ps atoms in the free volumes of the amorphous regions of the blends via pick-off annihilation. As a result of the addition of Nylon-6,6 to PVA, the *o*-Ps lifetime *τ*_3_, *o*-Ps intensity *I*_3_, and relative fractional of hole volume size *f*_r_ are depicted in Figure 8. The right axis of the upper figure represents the hole volume size *V*_f_ calculated using Equations (4) and (5). In the Nylon-6,6/PVA blends, the *o*-Ps lifetime *τ*_3_ was between that of pure PVA and pure Nylon-6,6. As the proportion of Nylon-6,6 blended with PVA rises, the *o*-Ps lifetime *τ*_3_ or hole volume size increases exponentially.

The change in free volume caused by an increase in the concentration of pure Nylon-6,6, which has a larger free-volume size (0.94 nm^3^) than pure PVA (0.046 nm^3^), or the emergence of new bonds could account for the rise in t_3_. The rise in t_3_ suggests that the PVA free-volume hole inner chain is expanding in some way. To put it another way, Nylon-6,6’s larger holes can be filled with PVA’s smaller free-volume holes. As can be seen in Figure 8, the *o*-Ps intensity *I*_3_ decreases as the concentration of Nylon-6,6 on the PVA/Nylon-6,6 blend increases. It is common knowledge that the polymer’s degree of crystallinity can affect the *o*-Ps intensity *I*_3_ [29]. A high crystallinity level results in a lower *I*_3_, while a low crystallinity level results in a high *I*_3_. The behaviour of the *o*-Ps intensity *I*_3_ (*I*_3_ for pure Nylon-6,6 < *I*_3_ for pure PVA) correlates with the high crystallinity of pure Nylon-6,6 in comparison to pure PVA. Moreover, it is clear from Figure 8 that, at low Nylon-6,6 loadings, additional free volume was added when Nylon-6,6 was blended with PVA, as seen in the relative fractional of the free-volume size *f*_r_ derived using Equation (9). This result indicates that the Nylon-6,6 interacted with the smaller free-volume distribution in PVA at the molecular level.

In a polymer blend, the free volume does not simply comprise the addition of each individual free volume; instead, the relative fractional of the free-volume hole of the blend can be expressed as [81]:*f*_r_ = A_1_
*f*_r1_ + A_2_ *f*_r2_ + A_1_ A_2_ Q (*f*_r1_ + *f*_r2_)^0.5^, (18)
where *f*_r1_ and *f*_r2_ are the relative fractional of the free-volume hole of polymer 1 (PVA) and polymer 2 (Nylon-6,6), respectively, and A_1_ and A_2_ are the volume fractions of the two polymers in the blend. Q is an interaction parameter that may be related to the interaction between dissimilar polymer chains [79]. Q > 0 indicates a positive deviation from the free-volume additivity, and Q < 0 indicates a negative deviation. The values of the interaction parameters Q are −0.057, −0.041, −0.043, and −0.068 for 20, 40, 60, and 80 wt.% of Nylon-6,6 concentration blended with PVA, respectively. The negative value of the interaction parameter Q indicates that blending PVA with Nylon-6,6 leads to a negative deviation. Therefore, the change in Q is possibly due to the interaction between the Nylon-6,6 and PVA polymer chains, enhanced by the interfacial agent. The present data is similar to the data of Wang et al. [82], who revealed that in a polypropylene/ethylene-propylene-diene monomer blend (PP/EPDM), the free volume shows a negative deviation due to the phase interaction. Sood et al. [83] observed negative interaction parameters in a series of miscible polymer blends through the measurement of viscosity. Liu et al. [84] also found that the interaction parameter is negative in miscible blends while it is positive in immiscible blends.

All these results reflect that the compatibilizer had a strong effect on the free-volume properties in the blend. However, the information about the free volume that we obtained from the finite term analysis of the positron lifetime spectra is limited because it gives only a few averaged lifetimes. In other words, what we obtained from this analysis is the average free-volume size. However, most of the time, the free-volume size has a distribution, and obtaining this information will be more helpful for us to analyse the detailed microstructural information of polymers and polymer blends. In the present work, the LT10 program [38] was used to analyse the measured spectra. The lifetime distribution data are shown in Figure 9. As shown in Figure 9A, in the pure and blended polymers, the *o*-Ps lifetime has one clear peak located in the range from 0.5 to 3.0 ns. The shorter lifetime peak is close to the *o*-Ps lifetime in PVA, and the longer lifetime peak is related to the *o*-Ps lifetime in Nylon-6,6. In addition, the *o*-Ps lifetime distributions for the blended samples were located between these peaks for the pure polymers. By increasing the Nylon-6,6 content in the blended PVA, the *o*-Ps distribution became narrower. The behaviours for the *o*-Ps hole radius [Figure 9B] and *o*-Ps hole volume size [Figure 9C] are similar to those of the *o*-Ps lifetime distribution. The one clear *o*-Ps lifetime peak (free-volume size peak) for the blended samples shows strong evidence of no phase separation. Because this blend is miscible, the dispersed Nylon-6,6 tends to have as small a domain size as possible in the interaction with PVA polymeric chains. Therefore, *o*-Ps will form and annihilate in one phase.

Understanding the polymer properties requires a decent comprehension of the free-volume hole. The free-volume hole is generally generated from the irregular dispersion of holes in polymer atomic chain fragments. The free-volume hole is responsible for the directional conduction of particles in polymers. Controlling the hole volume content of any polymer can change its physical and substance properties. Figure 10 shows the relationship between the hole volume size *V*_f_ and the DC ionic conductivity σ_DC_ for the blended polymer (Nylon-6,6/PVA). It is shown in Figure 10 that the ionic conductivity is found to decrease with an expansion in the hole volume size because ions are not permitted to take a leap toward an adjoining site when there are enormous enough free volumes between them [85]. Therefore, Figure 10 represents a strong correlation between the microscopic properties obtained from the PAL spectroscopy results and other macroscopic properties assessed from other measurements such as ionic conductivity in the data of Mohamed et al. [86]. These correlations indicate that these macroscopic properties, such as proton conductivity, are governed by the free-volume size.

## 5. Conclusions

The free-volume properties and electrical properties of Nylon-6,6/PVA blends were studied using positron annihilation lifetime spectroscopy and electrical measurements. An interaction between Nylon-6,6 and PVA chains was found in the blends and their ionomers. The activation energy of the thermal temperature for some blends is lower than that of pure polymers, indicating that their thermal stability is somewhere in between that of Nylon-6,6 and PVA. The degree of crystallinity of the blend is lower than that of the pure components due to the negative effects of each component. Due to an increase in the number of mobile charge carriers in the blend with a higher PVA concentration, the ionic conductivity increases as the concentration of Nylon-6,6 blended with PVA decreases. The hole volume size increases exponentially with increasing concentrations of Nylon-6,6 blended with PVA. The Nylon-6,6 interacted with the smaller free-volume distribution in PVA at the molecular level. There was no phase separation in the blended samples as deduced from the free-volume distribution of the Nylon-6,6/PVA blends. The negative value of the interaction parameter indicates that blending PVA with Nylon-6,6 leads to a negative deviation (miscible blends). The strong correlation between the free-volume size and other macroscopic properties such as ionic conductivity indicates that these macroscopic properties are controlled by the free-volume size.

## Figures and Tables

**Figure 1 polymers-15-00107-f001:**
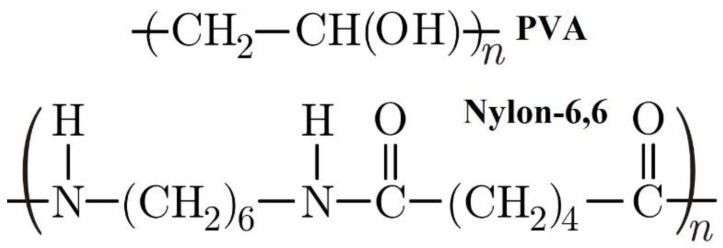
The chemical structure of PVA and Nylon-6,6 polymers.

**Figure 2 polymers-15-00107-f002:**
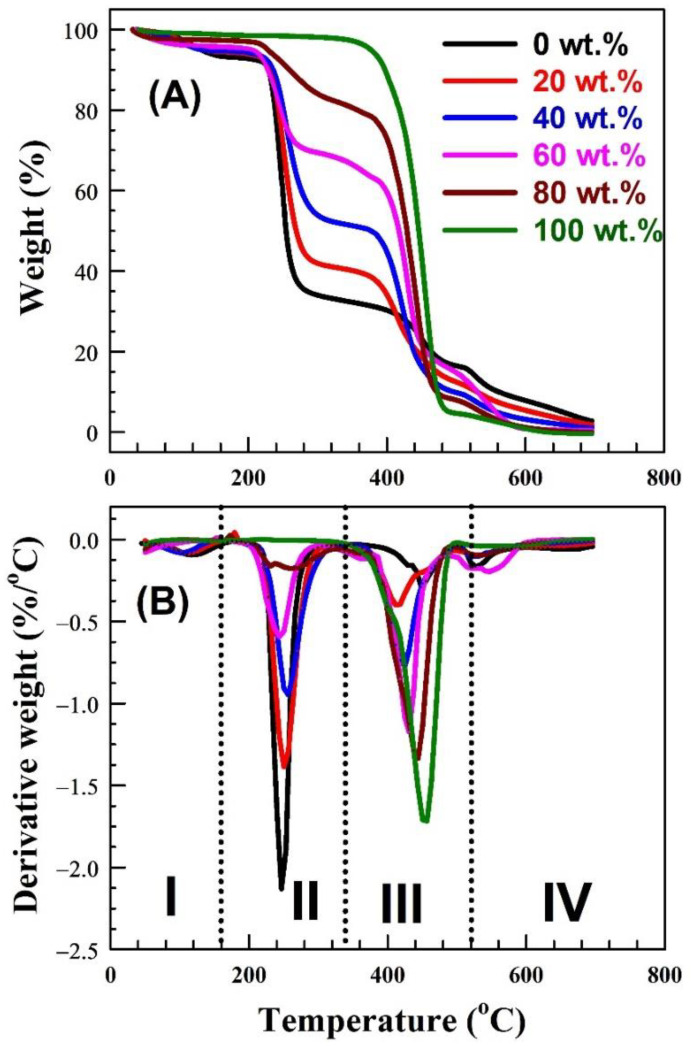
(**A**) Thermogravimetric analysis (TGA) and (**B**) the derivative thermogravimetric (DTG) for poly(vinyl alcohol) (PVA) blended with different concentrations of Nylon-6,6.

**Figure 3 polymers-15-00107-f003:**
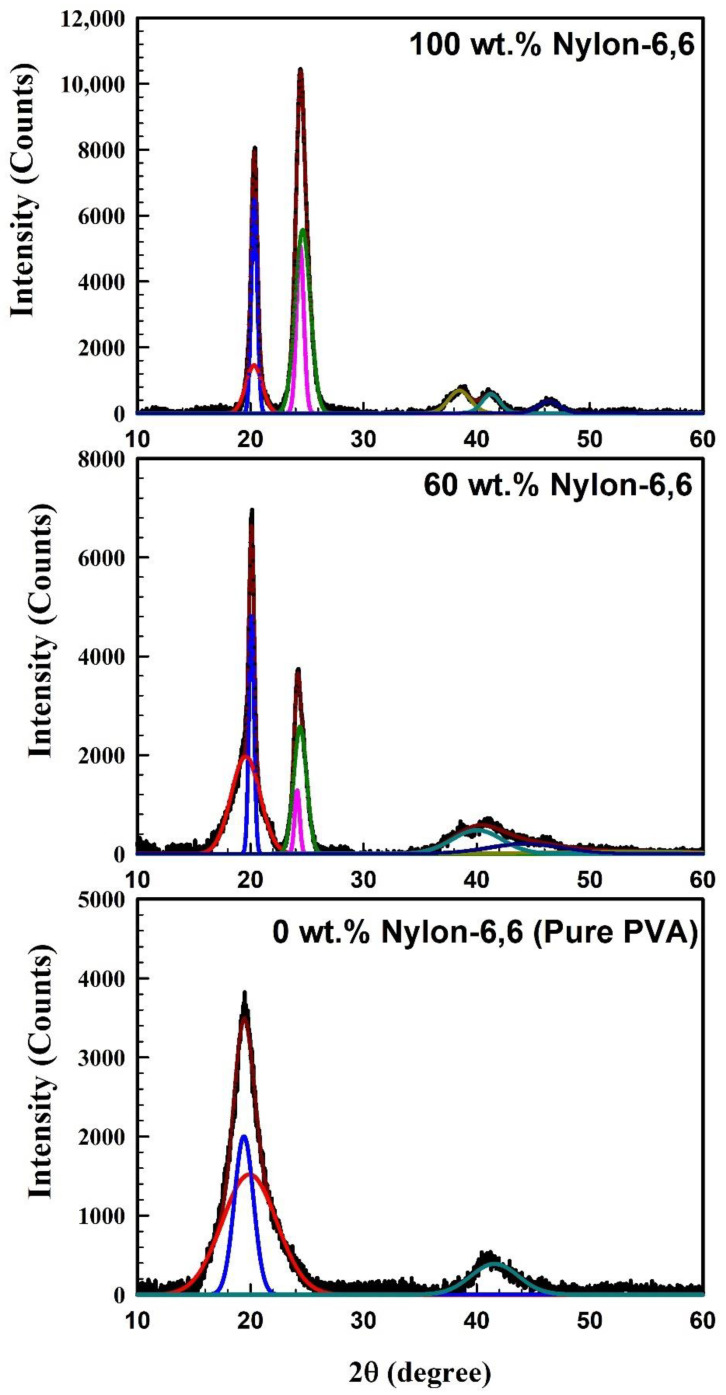
WAXD spectra for pure Nylon-6,6, polymer blend (60 wt.% Nylon-6,6), and pure PVA. The blue and pink peaks represent the crystalline peaks while the red and green peaks represent the amorphous peaks for the samples.

**Figure 4 polymers-15-00107-f004:**
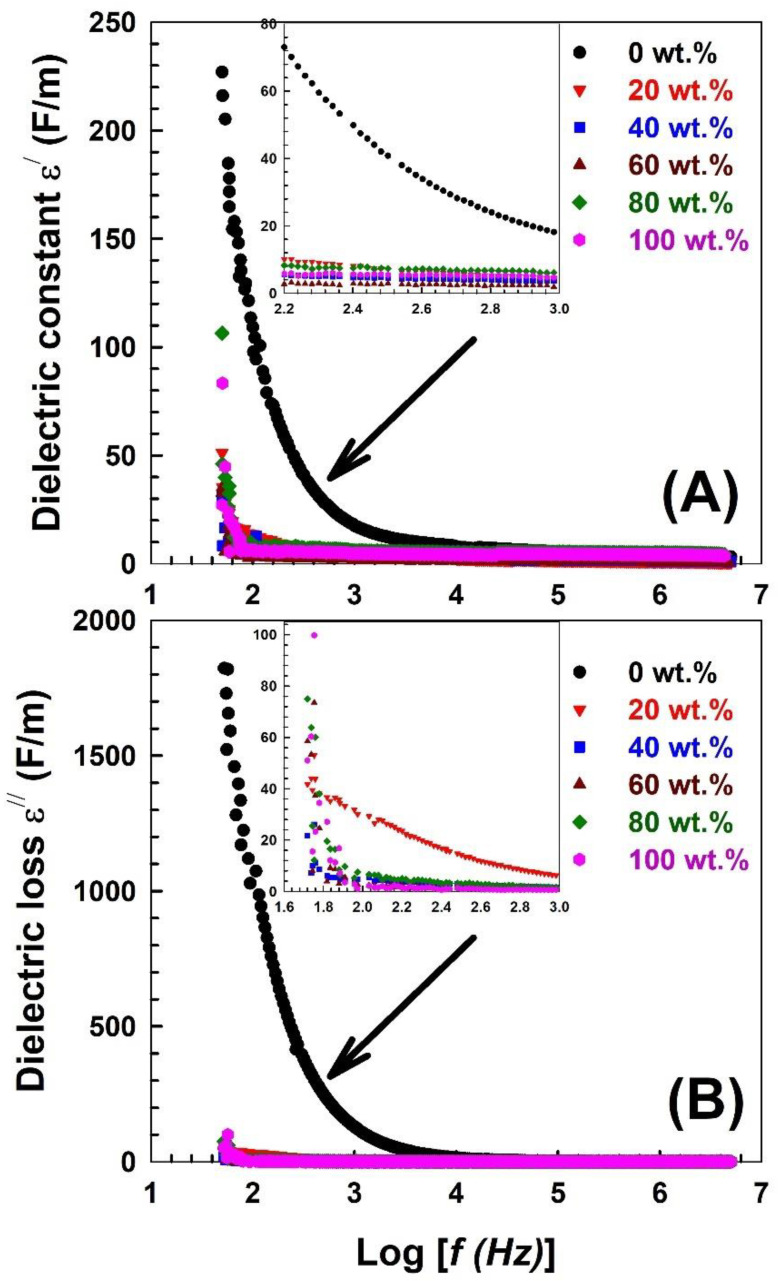
Frequency dependence of (**A**) the dielectric constant ε′ and (**B**) the dielectric loss ε′′ of the PVA blended with different concentrations of Nylon-6,6.

**Figure 5 polymers-15-00107-f005:**
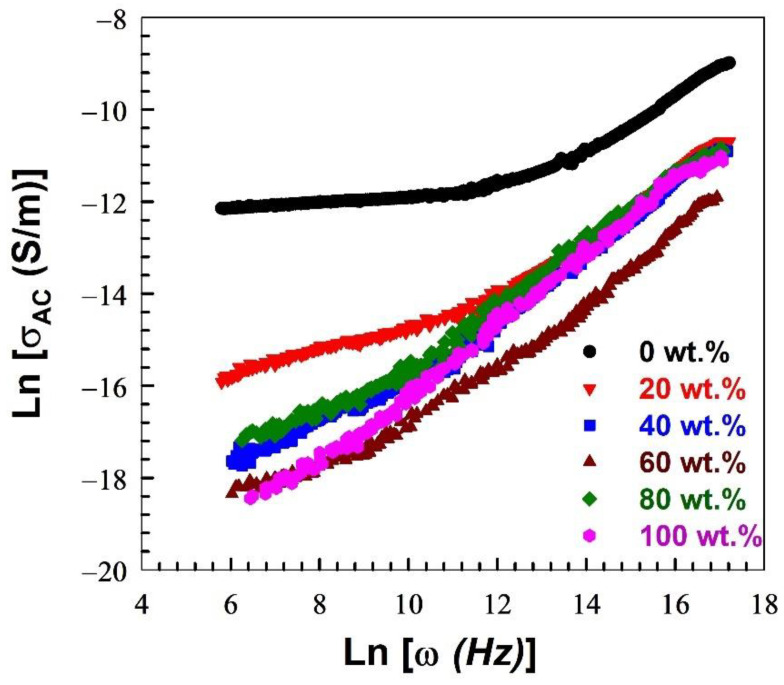
Variation of the ln (*σ_ac_*) with ln (*ω*) at various Nylon-6,6/PVA blend concentrations.

**Figure 6 polymers-15-00107-f006:**
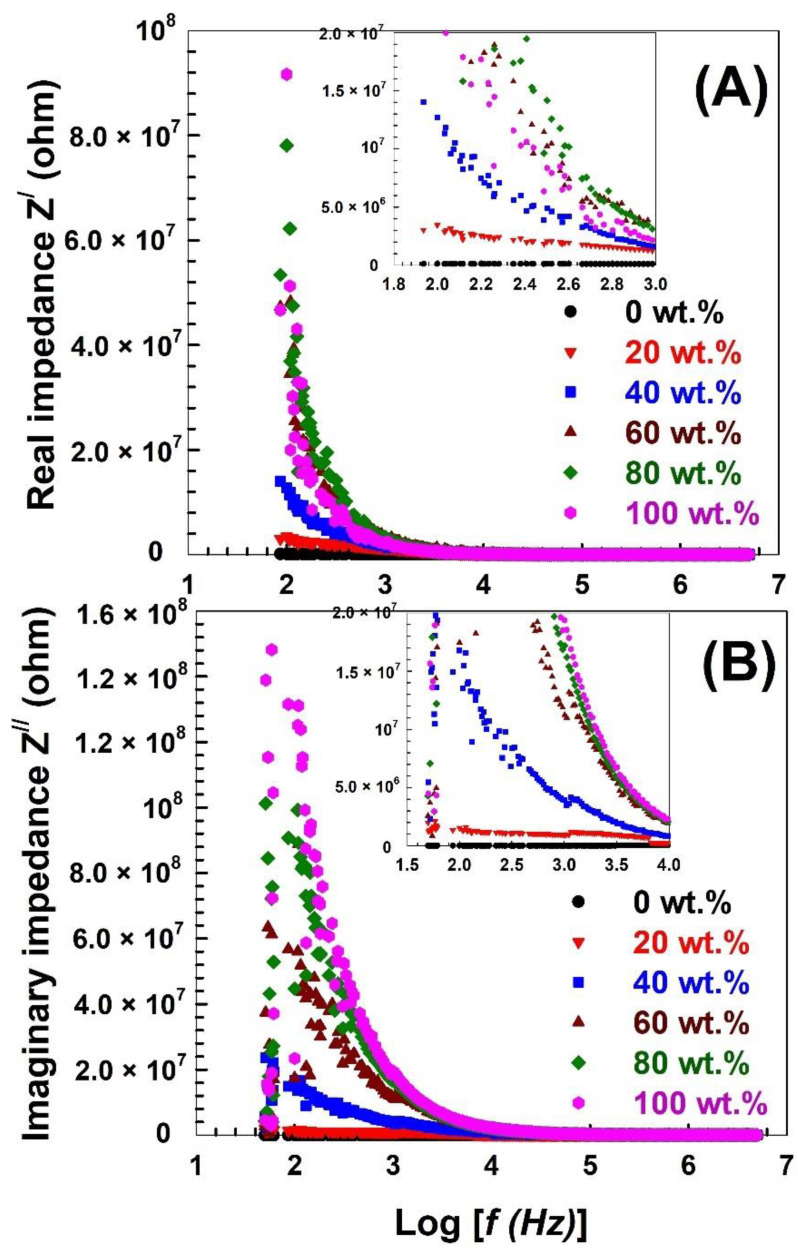
Frequency dependence of (**A**) real impedance Z^/^ and (**B**) imaginary impedance Z^//^.

**Figure 7 polymers-15-00107-f007:**
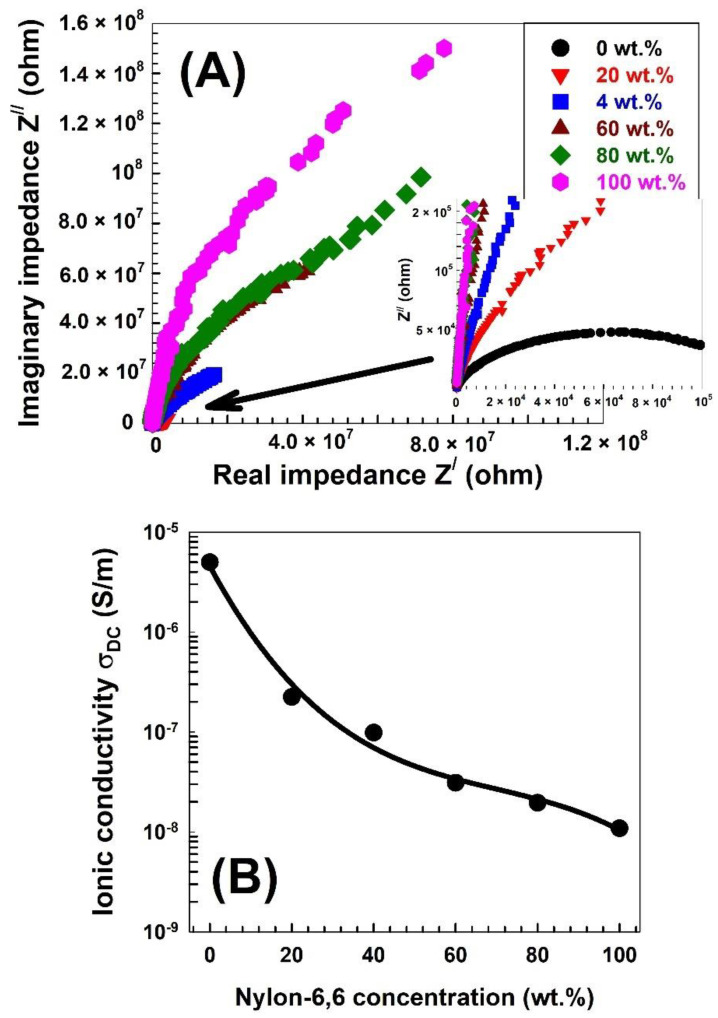
(**A**) Cole–Cole plot for the PVA blended with different concentrations of Nylon-6,6 and (**B**) the DC ionic conductivity σ_DC_ as a function of the Nylon-6,6 concentrations blended with PVA.

**Figure 8 polymers-15-00107-f008:**
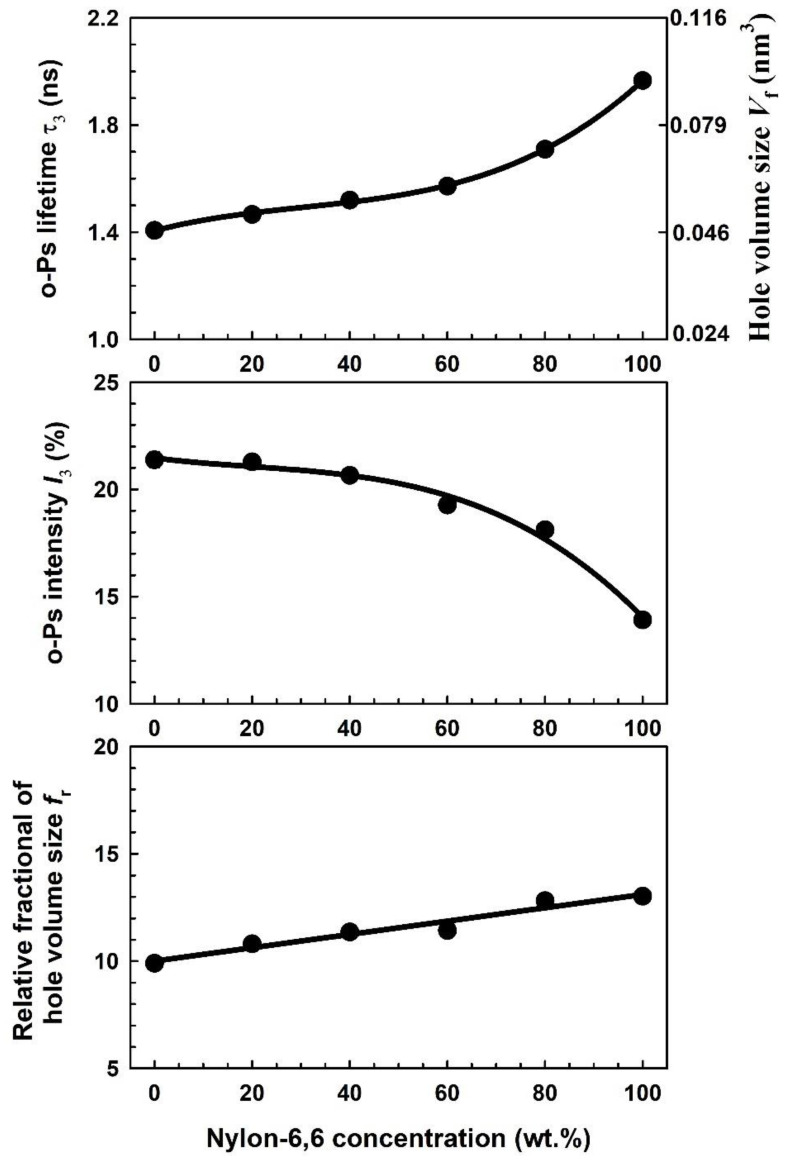
The *o*-Ps lifetime t_3_, its intensity *I*_3_, and relative fractional of hole volume *f*_r_ as a function of Nylon-6,6 concentrations blended with PVA. Included in the right top figure is the hole volume size *V*_f_ calculated using Equations (4) and (5).

**Figure 9 polymers-15-00107-f009:**
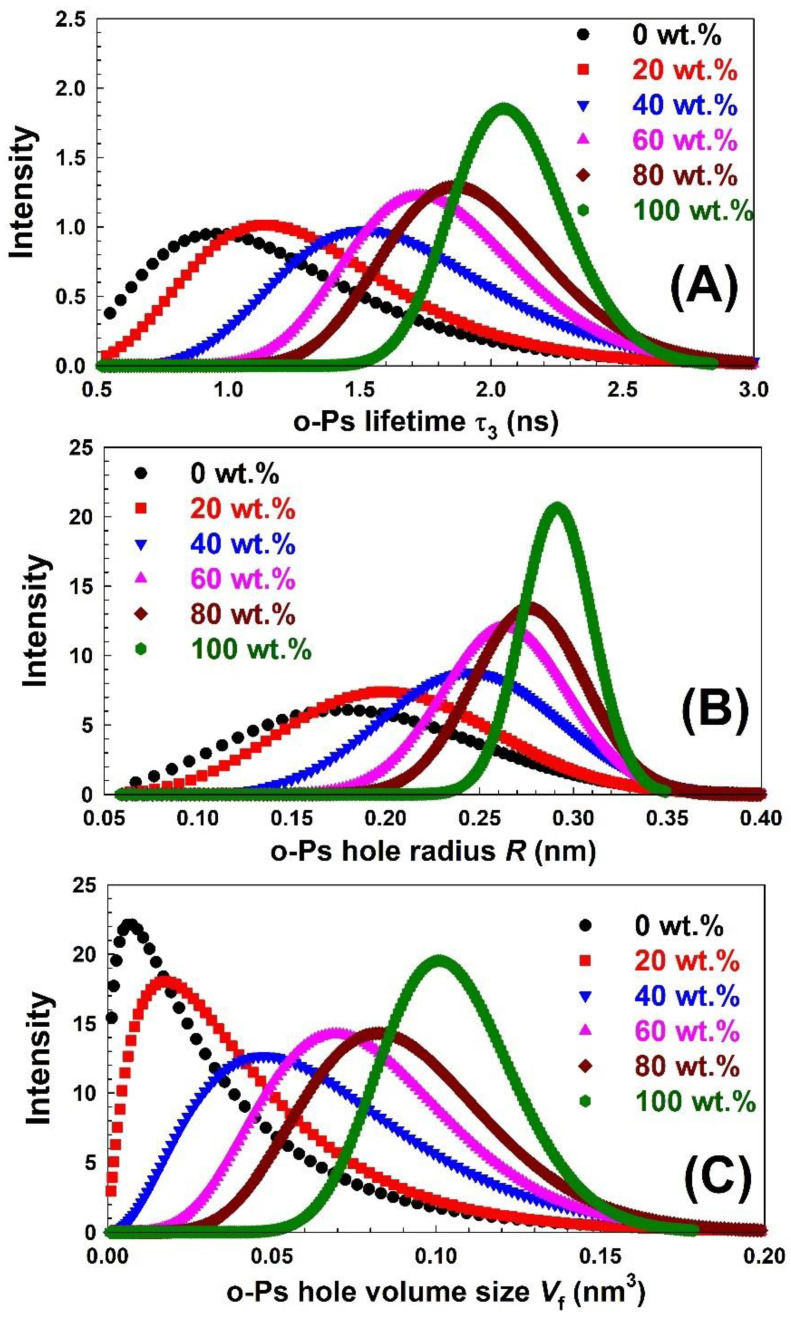
Distribution of (**A**) o-Ps lifetime *τ*_3_, (**B**) the *o*-Ps hole radius *R*, and (**C**) *o*-Ps hole volume size *V*_f_ for the PVA blended with different concentrations of Nylon-6,6 deduced using LT10 program.

**Figure 10 polymers-15-00107-f010:**
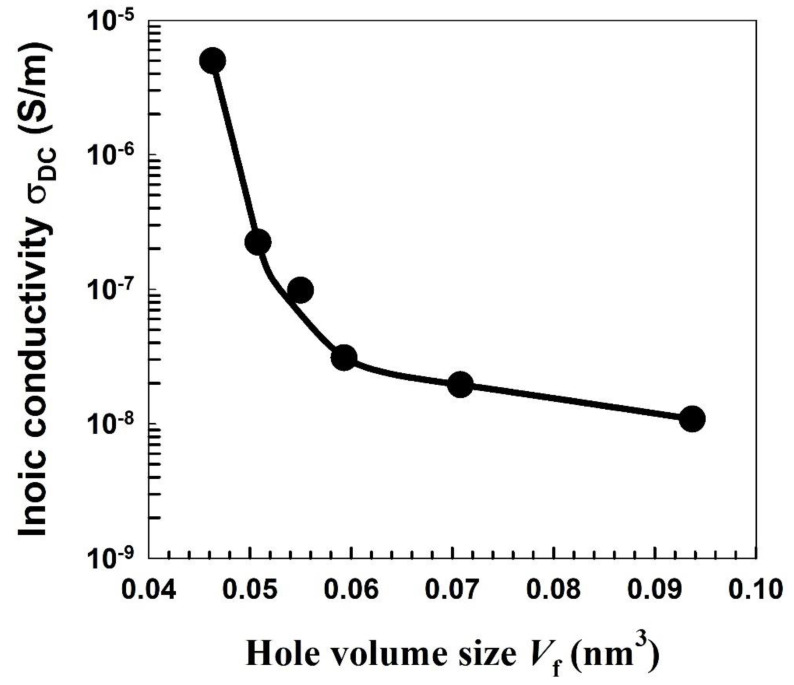
The correlation between the hole volume size *V*_f_ and the DC ionic conductivity σ_DC_ for the blend polymer (Nylon-6,6/PVA).

**Table 1 polymers-15-00107-t001:** The activation energy of the thermal decomposition *E*_b_ for the investigated blends at different decomposition temperatures.

Nylon-6,6 Concentration Blended with PVA (wt.%)	Activation Energy of PVA (J/Mole) At ~240 °C	Activation Energy of Nylon (J/Mole) At ~440 °C
0	2.1 × 10^5^	----
20	7.6 × 10^4^	2.73 × 10^5^
40	1.18 × 10^5^	2.94 × 10^5^
60	2.9 × 10^4^	8.2 × 10^4^
80	8.5 × 10^4^	2.3 × 10^5^
100	----	1.52 × 10^5^

**Table 2 polymers-15-00107-t002:** The peak position, FWHM, and grain size for the crystalline and amorphous peaks for pure Nylon-6,6, pure PVA, and blend polymer (60 wt.% Nylon-6,6/40 wt.% PVA).

Sample	Crystalline Peak			Amorphous Peak		
	Peak Position	FWHM	Grain Size (nm)	Peak Position	FWHM	Grain Size (nm)
Pure Nylon-6,6	20.341°	0.610°	13.82	20.319°	1.678°	5.03
	24.399°	0.762°	11.14	24.628°	1.587°	5.35
Pure PVA	19.427°	1.985°	4.24	19.909°	5.657°	1.49
Blend polymer 60 wt.% Nylon-6,6/40 wt.% PVA	20.099°	0.53°	15.88	19.577°	2.985°	2.82
	24.130°	0.527°	16.11	24.396°	1.347°	6.30

## Data Availability

The data that supports the findings of this study are available within the article.

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
