# Peer review of "Investigation of Transport Mechanism and Nanostructure of Nylon-6,6/PVA Blend Polymers"

_polymers, 2022, doi:10.3390/polym15010107_

Round 1
Reviewer 1 Report
Dear authors,
This manuscript reports the study of free volume properties and electrical properties of PVA/Nylon-6,6 blends of various concentrations of Nylon-6,6 using thermogravimetric analysis (TGA), wide-angle X-ray diffraction (WAXD), electrical conductivity measurements, and particularly positron annihilation (PAL) spectroscopy. The conclusions are solid and convincing, and I think the manuscript is suitable for the journal of Polymers after major revision.
Here are my comments:
#1: In Figure 1, the chemical structure of PVA is incorrect.
#2: For reference no. 34, the page number is incorrect.
#3: For Figures 2, 7, and 11, the order of the panel figures is incorrect.
#4: For the introduction of the technique of positron annihilation spectroscopy, I think there is unnecessary to use such many equations, and please simplify it a little bit if possible.
#5: For the WAXD spectra in Figure 3, for each single panel figure, besides the overlapped peaks, why are there also crystalline peaks (in blue and pink) and amorphous peaks (in red and green)? And how did you get them?
#6: The dielectric constant characterization has been shown in Figures 6 – 8, occupying a large part of this manuscript. Therefore, I think it might be of great help if you can add some related information in the Abstract, Introduction, as well as Materials and Methods.
#7: In line 365, I guess “polyamide” might be “Nylon-6,6”.
I am very glad to review this manuscript, and once again, I recommend this manuscript to be accepted by the journal of Polymers after major revision.
Sincerely,
xxx
Author Response
Reviewer 1
#1: In Figure 1, the chemical structure of PVA is incorrect.
Thank you so much for your correction. The figure was corrected and inserted in the revised manuscript.
#2: For reference no. 34, the page number is incorrect.
Thank you for your correction. The page number of the reference was corrected to be “258” as “Jean, Y.C.; Mallon, P.E.; Schrader, D.M. Principles and applications of positron and positronium chemistry. World Scientific; UK, 2003, pp.258“
#3: For Figures 2, 7, and 11, the order of the panel figures is incorrect.
Thank you so much for your comment. The panel order of Figures (2, 7, and 11) was revised in the revised manuscript.
#4: For the introduction of the technique of positron annihilation spectroscopy, I think there is unnecessary to use such many equations, and please simplify it a little bit if possible.
Thank you so much for your comment. Equations 4 and 5 were deleted from the Introduction section.
#5: For the WAXD spectra in Figure 3, for each single panel figure, besides the overlapped peaks, why are there also crystalline peaks (in blue and pink) and amorphous peaks (in red and green)? And how did you get them?
Thank you so much for your comment. The obtained peaks from the WAXD analysis are only the main peaks (black ones) as in the next figure (for example). Such peaks with this broadening in its base indicate that a mix structure (crystalline and amorphous) exist in the samples. Fityk software has been used to deconvolute these peaks and show the sub-peaks as it is shown in the manuscript. It is well known that the crystalline peak had the smallest FWHM compared with the amorphous peak. The WAXD pattern of PVA, especially the peaks at about 2q = 19.5 oC included two peaks; the crystalline peak which has the smaller FWHM, and the other one is an amorphous peak which has the largest FWHM. A similar thing happens on the WAXD pattern for the Nylon-6,6 polymers.
#6: The dielectric constant characterization has been shown in Figures 6 – 8, occupying a large part of this manuscript. Therefore, I think it might be of great help if you can add some related information in the Abstract, Introduction, as well as Materials and Methods.
Thank you so much for your comment. Some information about the dielectric constant characterization was added to the abstract secretin as:”The dielectric properties of the blended samples were investigated for different frequencies (50 Hz- 5 MHz). σac versus frequency was found to obey Jonscher's universal power law. The calculated values of the s-parameter were increased from 0.53 to 0.783 for 0 and 100 wt.% Nylon6-6, respectively, and less than 1 indicated the hopping conduction mechanism. The barrier height (Wm) was found to increase from 0.33 to 0.72 for 0 and 100 wt.% Nylon6-6, respectively.”
Also, a paragraph was added in the Introduction section as : “The studies of the dielectric complex, the conduction mechanism, and the barrier height for the blended samples, in a wide range of frequencies (50 Hz to 5 MHz) have been carried out to investigate the dielectric constant of the samples which reveals the polarization in the samples and their ability to store electrical energy, and to identify the type of the conduction mechanism. The conduction mechanism of PVA/TiO2 was investigated by Yasmin K. et al. [17], and the correlated barrier hopping mechanism was the major conduction mechanism.“
In addition, some information was added to the Materials and Methods section as:” LCR meter (Hioki 3532, Japan) was used to measure the many parameters such as impedance, parallel-equivalent static capacitance, loss coefficient, and conductance for all the samples. Using some of these parameters, the dielectric properties, and complex impedance were calculated as a function of frequencies from 50 Hz to 5MHz. The measurements were done under a vacuum at 25 °C. The out-of-plane value of ionic conductivity sAC can be acquired using the conductance G as [47]”
#7: In line 365, I guess “polyamide” might be “Nylon-6,6”.
Thank you for your comment. “polyamide” was changed to Nylon-6,6 on the manuscript.
Reviewer 2 Report
In this manuscript, polymer composites based on poly(vinyl alcohol) (PVA) with different concentrations of Nylon-6, 6 were prepared. The thermal activation energy, e degree of crystallinity, and ionic conductivity were studied. I could recommend the publication of this manuscript on Polymers after a minor revision. The following questions and issues should be addressed:
1. The conceptual advances and significant impact of this manuscript should be highlighted in abstract.
2. The author should merge appropriately the figures.
3. The mechanical properties of PVA/Nylon-6,6 composite should be provided.
4. The thermogravimetric stability of PVA/Nylon-6,6 composite was enhanced as the concentrations of Nylon-6,6 increasing. The author should give a deep analysis and cite some related papers such as “Chemical Engineering Journal, 2022, 448, 137742; Chemical Engineering Journal, 2023, 451, 138882; ACS Applied Materials & Interfaces, 2021, 13, 55020-55028.”
Author Response
Reviewer 2
1. The conceptual advances and significant impact of this manuscript should be highlighted in abstract.
Thank you so much for your comment. Some information was added to the abstract section such as: ”The dielectric properties of the blended samples were investigated for different frequencies (50 Hz- 5 MHz). σac versus frequency was found to obey Jonscher's universal power law. The calculated values of the s-parameter were increased from 0.53 to 0.783 for 0 and 100 wt.% Nylon6-6, respectively, and less than 1 indicating the hopping conduction mechanism. The barrier height (Wm) was found to increase from 0.33 to 0.72 for 0 and 100 wt.% Nylon6-6, respectively.”
2. The author should merge appropriately the figures.
Thank you so much for your comments. The figures were merged to be 10 figures instead of 12.
3. The mechanical properties of PVA/Nylon-6,6 composite should be provided.
Thank you so much for your comment. You are completely right but unfortunately; it is not possible to do this measurement
4. The thermogravimetric stability of PVA/Nylon-6,6 composite was enhanced as the concentrations of Nylon-6,6 increasing. The author should give a deep analysis and cite some related papers such as “Chemical Engineering Journal, 2022, 448, 137742; Chemical Engineering Journal, 2023, 451, 138882; ACS Applied Materials & Interfaces, 2021, 13, 55020-55028.”
Thank you so much for your comment. Some information was added in the first paragraph of the Results and Discussion Section as: “The samples can be thermally degraded for different reasons, such as losing the adsorbed water, elimination of any equivalent groups or hydrogen atoms, or maybe disposal of carbon by converting it into carbon dioxide, as was introduced in the recently published studies [52-54].”
Round 2
Reviewer 1 Report
Dear Authors,
Thanks a lot for your so careful revision of your manuscript. And I am satisfied with your revision and answers. Therefore, I would suggest the Editor to accept it in present form.
Sincerely,
Xxx
Reviewer 2 Report
The authors have addressed my concerns. Thus it could be published now.